# Validation of a simplified risk prediction model using a cloud based critical care registry in a lower-middle income country

**Bharath Kumar Tirupakuzhi Vijayaraghavan**[1☯‡*], **Dilanthi Priyadarshini**[2☯‡],
**Aasiyah Rashan**[2], **Abi Beane**[3], **Ramesh Venkataraman**[1], **Nagarajan Ramakrishnan**[1],
**Rashan Haniffa**[3], **the Indian Registry of IntenSive care(IRIS) collaborators**[¶]

1 Department of Critical Care Medicine, Apollo Hospitals, India and Chennai Critical Care Consultants, Chennai, India, 2 Network for Improving Critical Care Systems and Training, Colombo, Sri Lanka, 3 Mahidol Oxford Tropical Research Unit, Thailand, Bangkok

☯ These authors contributed equally to this work.
‡ These authors share first authorship on this work.
¶ Membership of the Indian Registry of IntenSive care(IRIS) collaborators is listed in the Acknowledgments.
* bharath@icuconsultants.com

**Data Availability Statement:** Pooled data from IRIS are available from the IRIS Dashboard at https://nicst.com/picu-iris-public/. The IRIS

## Abstract

### Background

The use of severity of illness scoring systems such as the Acute Physiology and Chronic Health Evaluation in lower-middle income settings comes with important limitations, primarily due to data burden, missingness of key variables and lack of resources. To overcome these challenges, in Asia, a simplified model, designated as e-TropICS was previously developed. We sought to externally validate this model using data from a multi-centre critical care registry in India.

### Methods

Seven ICUs from the Indian Registry of IntenSive care(IRIS) contributed data to this study. Patients > 18 years of age with an ICU length of stay > 6 hours were included. Data including age, gender, co-morbidity, diagnostic category, type of admission, vital signs, laboratory measurements and outcomes were collected for all admissions. e-TropICS was calculated as per original methods. The area under the receiver operator characteristic curve was used to express the model's power to discriminate between survivors and non-survivors. For all tests of significance, a 2-sided P less than or equal to 0.05 was considered to be significant. AUROC values were considered poor when ≤ to 0.70, adequate between 0.71 to 0.80, good between 0.81 to 0.90, and excellent at 0.91 or higher. Calibration was assessed using Hosmer-Lemeshow C -statistic.

### Results

We included data from 2062 consecutive patient episodes. The median age of the cohort was 60 and predominantly male (n = 1350, 65.47%). Mechanical Ventilation and vasopressors were administered at admission in 504 (24.44%) and 423 (20.51%) patients

collaboration supports and welcome data sharing. Our agreement with participating sites in the registry is only for the sharing of deidentified data between them and the registry coordinating centre for the purposes of audit, quality improvement and specific research questions. We are not allowed to post data on a repository or any other public database. Raw data will be made available to qualified researchers who provide a detailed and methodologically sound proposal with specific aims that are clearly outlined. Such proposals will be screened by the IRIS Steering committee for approval. Data sharing will be for the purposes of medical research and under the auspices of the consent under which the data were originally gathered. To gain access, qualified researchers will need to sign a data sharing and access agreement and will need to confirm that data will only be used for the agreed upon purpose for which data access was granted. Researchers can contact the corresponding author through electronic mail (bharath@icuconsultants.com) for such access; alternatively, IRIS can be contacted at info@irisicuregistry.org and joinus@irisicuregistry. org.

**Funding:** Funding: This work is partially supported by the Wellcome Trust (https://wellcome.ac.uk/ what-we-do/our-work/innovations-flagships) and Mahidol Oxford Tropical Research Unit (https:// www.tropmedres.ac/) Authors RH and AB were co-applicants on the wellcome innovations grant WT215522/Z/19/Z Role of funder: The funder had no role in the design, conduct, analysis or decision to submit for publication.

**Competing interests:** The authors declare no competing interests.

respectively. Overall, mortality at ICU discharge was 10.28% (n = 212). Discrimination (AUC) for the e-TropICS model was 0.83 (95% CI 0.812–0.839) with an HL C statistic p value of < 0.05. The best sensitivity and specificity (84% and 72% respectively) were achieved with the model at an optimal cut-off for probability of 0.29.

## Conclusion

e-TropICS has utility in the care of critically unwell patients in the South Asia region with good discriminative capacity. Further refinement of calibration in larger datasets from India and across the South-East Asia region will help in improving model performance.

## Introduction

Severity of illness scoring systems such as the Simple Acute Physiology Score (SAPS) [1] and the Acute Physiology and Chronic Health Evaluation (APACHE) [2] help in risk prediction, benchmarking, quality improvement and patient selection for research. Over the past three decades, several iterations of these models have been developed and validated based on changes in the epidemiology of critical illness and substantial improvements in survival [3, 4]. Most of these models have been evaluated in the context of high-income countries (HICs). There are several limitations to the use of these models in middle income and lower-middle income countries (LMICs) such as differences in epidemiology of critical illness, including the high burden of tropical infections, the lack of resources for data collection, burden of data collection due to the large number of variables, missing variables and the absence of electronic health records that would otherwise facilitate seamless data flow [5]. To overcome these problems, researchers have developed and tested newer simplified models in LMICs [6, 7].

Recently, in Asia, one such simplified model, the e-TropICS (described as 'model 1' in the original manuscript) has been developed and validated [8]. The researchers, cognizant of the limited availability of variables and consequent high proportion of missingness, which has hampered intensive care units(ICUs) in resource limited settings from being able to utilise existing prognostic scores, aimed to develop a comparable score based on a more universally obtainable set of variables. However, this score has not been externally validated. We, therefore, aimed to validate the performance of the e-TropICS model on a multi-centre Indian data set from the recently established Indian Registry of IntenSive care (IRIS) [9].

## Methods

### Study setting

Seven ICUs located in 6 private and 1 not-for-profit institution, part of the Indian Registry of IntenSive care(IRIS) contributed data to this study. Of these, 5 were general (mixed medical-surgical) ICUs and two were medical ICUs. None of the participating ICUs from this study contributed data for the original model development and were only involved in this external validation exercise.

The Indian Registry of IntenSive care, a cloud-based registry of critical care units was established in Jan 2019 [9]. Details of the implementation and preliminary results of the case-mix program have been previously published [9].

## Patients

All patients reported to the registry between January 2019 to May 2019, were considered. Patients > 18 years of age with an ICU length of stay > 6 hours were included in the study. Patients with missing outcomes and those not meeting the inclusion criteria were excluded

## Data collection

This retrospective study used data collected as part of the IRIS dataset. Age, gender, pre-existing co-morbidity, diagnostic category, type of admission (planned, unplanned, medical or surgical), physiological vital signs and laboratory measurements were collected as per the definitions described for e-TropICS (Table 1) for all consecutive admissions. ICU outcomes rather than hospital outcomes were collected due to well-described logistical challenges in such settings [5, 8]. Data was collected daily by either nursing staff or by data collectors appointed to the registry network, all of whom had been trained in the process of data acquisition. Daily telephone reminders encouraging data input and checks for consistency of the number of admissions, discharges and outcomes from each ICU were undertaken by staff from the central coordinating centre. In-built measures in the data entry portal such as mandatory fields, range validations, drop down and checkboxes as opposed to free text entries were employed to promote fidelity of data recording.

## Ethics and consent

The study was approved by the Institutional Ethics Committee centrally at the study coordinating centre (Institutional Ethics Committee, Apollo Main Hospital- AMH-021/07-19). The informed consent model used in the registry has been described and published previously [9]. Briefly, participating sites in the registry either modified their general intensive care consent to include a clause on deidentified data collection or had waiver of the individual consent requirement.

## Statistical analysis

Availability of physiological and laboratory measurements was described using descriptive statistics. e-TropICS was calculated as per the authors' original methods [8]. The area under the

**Table 1. Patient characteristics at the time of ICU presentation.**

| Characteristic n = 2062 | Study dataset | | | Data from original model (Haniff et al.)[8] | |
|---|---|---|---|---|---|
| | All (n = 2062) | Dead (n = 212,10.3%) | Alive (n = 1850, 89.7%) | Dead n = 1031 | Alive N = 2590 |
| Planned admission (n (%)) | 383(18.57) | 31(14.62) | 352(19.03) | - | - |
| Gender male (n (%)) | 1350(65.47) | 136(64.15) | 1214(65.62) | - | - |
| LoS(mean, SD) | 3.41(4.20) | 3.22(3.91) | 5.34(6.34) | - | - |
| Age(Median, range) | 60(18–110) | 64(22–92) | 60 (18–110) | 54(16–102) | 56(16–103) |
| Heart rate(mean, SD) | 94.35(22.95) | 104.50(26.59) | 93.18(22.21) | 109(24) | 100(24) |
| GCS(Median, range) | 14(3–14) | 10(3–14) | 14(3–14) | 9(2–15) | 15(2–15) |
| Respiratory rate(mean, SD) | 22.83(5.58) | 25.70(8.06) | 22.50(5.12) | 24(8) | 23(6) |
| Systolic BP(mean, SD) | 129.20(25.91) | 120.53(29.14) | 130.19(25.33) | 132(35) | 139(29) |
| Blood urea(median, range) | 35(0.8–400) | 58.5(11–239) | 33(0.8–400) | 59(0.6–411) | 32(0.9–672) |
| Haemoglobin, g/dL(mean, SD) | 11.46(2.77) | 10.34(2.62) | 11.59(2.76) | 10.8(2.8) | 11.6(2.4) |
| Vasopressor used on admission (n (%)) | 423(20.51) | 124(58.49) | 299(16.16) | 317(35.7) | 177(8.7) |
| Mechanical ventilation on admission (n (%)) | 504(24.44) | 122(57.55) | 382(20.65) | 899(87) | 2054(79) |

receiver operator characteristic curve (AUROC) was used to express the model's power to discriminate between survivors and non-survivors. For all tests of significance, a 2-sided $P$ less than or equal to 0.05 was considered to be significant. AUROC values were considered poor when less than or equal to 0.70, adequate between 0.71 to 0.80, good between 0.81 to 0.90, and excellent at 0.91 or higher [10]. Calibration for the model was assessed using Hosmer-Lemeshow C -statistic and higher values of the Hosmer-Lemeshow C-Statistic indicate poorer calibration.

All analysis was performed using Stata software version 13.1 [11].

**Handling of missing data and analysis.** When faced with high proportions of missing data, one approach is to assume normality for a variable when not measured or unavailable, resulting in a score of "0" in weighted scoring systems. Such an approach may not be justified in LMICs where measurements may be unavailable due to lack of resource availability or to differing approaches in decision-making in critical illness. Assumptions of normality in the above manner can adversely impact model performance by underestimating severity scores. In this study, multiple Imputation (MI) with chained equations was employed to handle missing data. It was assumed that the missingness of a variable depends on some of the other observed variables i.e. Missing At Random (MAR). MI was performed using sequential imputation using chained equations. This is a multivariate approach that allows the flexibility of modelling different types of data within the same model with different rules being chosen based on type of data(predictive mean matching for continuous, logit for categorical and so on. The number of imputations (M) was set at 20 and "k-nearest neighbours" (kNN#, Stata syntax) was set at 10. Multiple Imputation (MI) generates several values reflecting the uncertainty in the estimation of the imputed value. The scores (and their mortality probabilities) were then calculated individually for each of the 20 multiple imputed datasets. The mean of 20 probabilities was then calculated and used the MI mortality prediction. As a secondary method, we also performed a complete case analysis and report the AUROC.

## Results

### Characteristics of population and availability of variables for the e-TropICS model

During the evaluation period, 2094 consecutive patient episodes were reported to the IRIS registry from the seven participating centres. Thirty-two patients were excluded and for the final analysis 2062 patients were included; 19 patients were less than 18 years of age and 13 had no outcome information.

The characteristics, demographics and outcomes for these episodes is described in Table 1. The median age of the cohort was 60 and predominantly male (n = 1350, 65.47%). Planned admissions accounted for 383 (18.57%) episodes. Mechanical Ventilation and vasopressors were administered at admission in 504 (24.44%) and 423 (20.51%) patients respectively. Overall, mortality at ICU discharge was 10.28% (n = 212). S1 Fig presents the commonest APACHE II diagnostic categories and the corresponding ICU outcome information.

Availability of the variables for the e-TropICS model is described in Table 2. Availability was lowest for blood urea (88.60%) and highest for gender and admission type (100%). For all other variables, availability ranged from 95% to 100%.

### Ability of models to predict mortality

Discrimination (AUC) for the e-TropICS model was 0.83 (95% CI 0.81–0.84) (Table 3, Fig 1) with an HL C statistic p value of < 0.05 suggesting poor model fit. The best sensitivity and

**Table 2. Availability of the parameters of e-Tropics model.**

| Parameters | Availability out of 2062, n(%) |
| --- | --- |
| Eye_opening | 2056(99.71) |
| Motor_response | 2057(99.76) |
| Verbal_response | 2056(99.71) |
| GCS | 2056(99.71) |
| Mechanically ventilated (yes/no at admission) | 2052(99.52) |
| Vasoactive drugs (yes/no at admission) | 2050(99.42) |
| Systolic BP | 2057(99.76) |
| Respiratory rate | 2056(99.71) |
| Haemoglobin | 2013(97.62) |
| Blood urea | 1827(88.60) |

specificity (84% and 72% respectively) were achieved with the model at an optimal cut-off for probability of 0.29. Fig 2 provides the calibration plot of the expected probability vs. the observed probability.

The results of the complete case analysis are presented in S2 Table and S2 Fig.

## Discussion

Our study validates the performance of a simplified prognostic model designed for use in critical care units, where information needed to calculate prognostic models from HIC may be absent or burdensome. In this multi-centre cohort from the recently established IRIS critical care registry, e-TropICS had good ability to discriminate death, but poor calibration. This suggests that in this cohort, the model can identify those patients at greatest risk of death, but has less ability to differentiate between degrees of severity of illness.

Several prognostic models for critically ill adults are validated and in use in HIC healthcare systems. However, their applicability in settings where due to limited point of care testing, data collection resources and even perhaps appropriate judicious use of laboratory tests, remains limited. Whilst missingness can be managed for the purposes of performance assessment and validation, a score which is not easily calculated—has limited application in the clinical setting. The e-TropICS model, developed from a South Asian dataset attempts to overcome several of these challenges by limiting model covariates to clinical information that would likely be requested by clinicians, and that are likely available in all health systems. Availability of data for validation was much higher than reported in previous literature from similar middle-income settings and ranged from 95–100%. Only blood urea had an availability less than 95%. Of note, the decision to perform laboratory tests is influenced by clinician preference, in addition to access to equipment, disposables, costs and expertise.

**Table 3. Performance of the e-tropICS model with multiple imputation.**

| Performance item | MI model N = 2062 |
| --- | --- |
| Probability, mean (SD) | 0.29(0.003) |
| Optimal cut-off probability | 0.23 |
| Sensitivity (at optimum cut-off) | 0.84 |
| Specificity (at optimum cut-off) | 0.72 |
| AUC (95% CI) | 0.83(0.812–0.839) |
| H/L C-statistic (p) | 340.87(0.00) |
| Brier score (95% CI) | 0.12(0.120–0.127) |

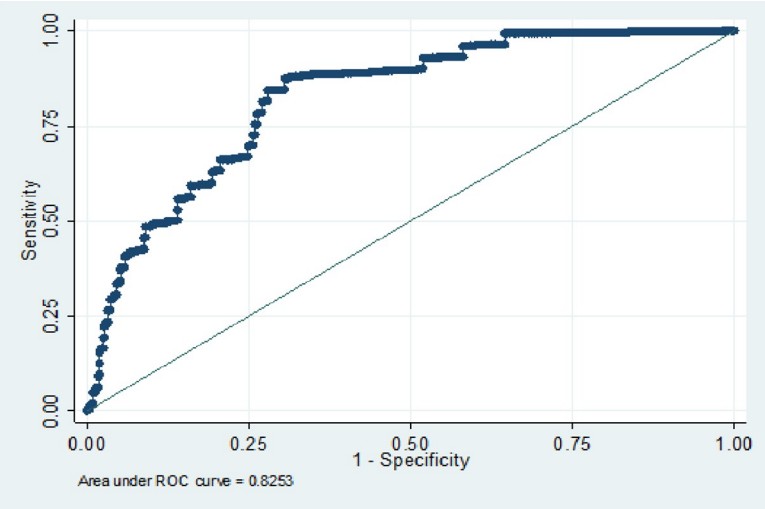

**Fig 1. ROC curve for imputed model.**

e-TropICS model had good discriminative ability (AUC of 0.81), but poor calibration. This is not surprising though, as several well-established prediction models, when validated externally have shown poor calibration [12]. Several reasons could explain this including the limitations of the HL test itself such as a high sensitivity to the sample size [13, 14]. Other reasons could include differences in case-mix [15]. Another potential explanation is the lower mortality in our dataset as compared to the dataset from which e-TropICS was developed (10.2% versus 28.4%). Previous research has shown that even small differences in mortality can affect the calibration of a model [16]. Whilst both good discrimination and good calibration are desirable for prognostic scores to be deemed suitable for application in clinical practice, in reality,

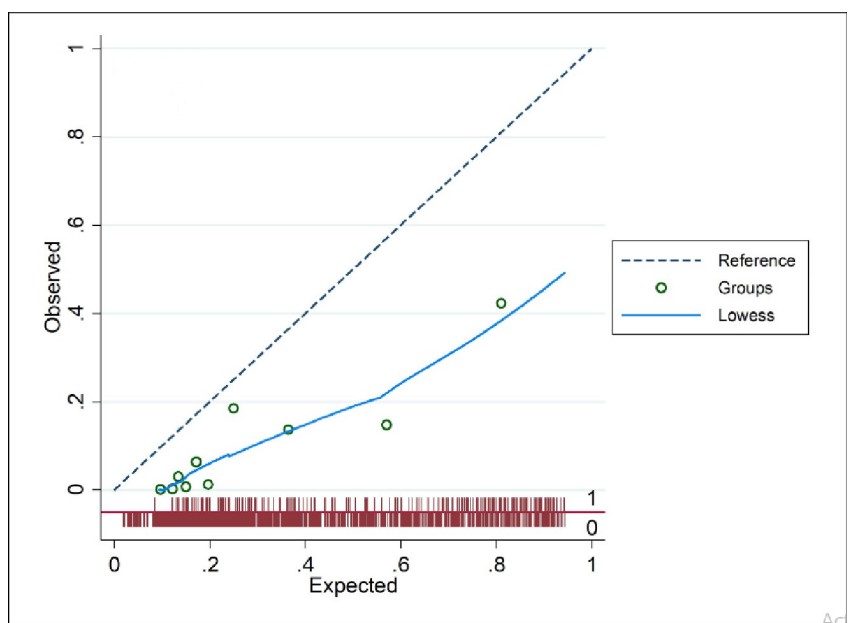

**Fig 2. Calibration plot of the expected probability (deciles) versus the observed probability.**

once an appropriate and implementable model is selected for use in a clinical setting, it can and perhaps should be regularly calibrated for the current population to which it is being applied. Our group is undertaking model refinement in the IRIS and in the recently established nine country critical care registry collaboration (Crit Care Asia) in South and Southeast Asia.

Our study has also demonstrated the feasibility of using registry data in a resource-limited setting to validate a locally relevant model. There remains limited investment in systems that enable routine data collection in LMICs and subsequently efforts to benchmark services and evaluate interventions to improve care remain hampered by low availability of information. The greater availability reported in this validation compared to earlier studies in the region suggests that investment in health system strengthening including the implementation of critical care registries can improve the availability of information during critical care admission.

## Conclusion

e-TropICS has utility in the care of critically unwell patients in the South Asia region. e-TropICS thus offers a prediction model that is simplified with low data collection burden for resource-limited settings. Further refinement of calibration of this model in larger datasets from India and across the South-East Asia region will help in improving model performance.

## Supporting information

**S1 Fig. Commonest APACHE II diagnostic categories and ICU outcomes.**
(TIF)

**S2 Fig. ROC curve for complete case analysis.**
(TIF)

**S1 Table. Variation of sensitivity and specificity at different cutpoints.**
(DOCX)

**S2 Table. Performance of the e-TropICS model (complete case analysis).**
(DOCX)

**S3 Table. Number of patients included from each site.**
(DOCX)

## Acknowledgments

IRIS collaborators:

1. Dr Devachandran Jayakumar and Dr Pratheema Ramachandran: Apollo Specialty Hospital, OMR, Chennai, India

2. Dr Deedipiya Devaprasad and Dr Vijay Chakravarthy: Apollo Specialty Hospital, Teynampet, Chennai, India

3. Dr Ashwin Mani and Dr Meghena Mathew: Apollo First Med Hospital, Kilpauk, Chennai, India

4. Dr Ebenezer Rabindrarajan and Dr Usha Rani: Apollo Specialty Hospital, Vanagaram, Chennai, India

5. Dr Niyaz Channanath Ashraf: IQRAA Hospital, Calicut, India

6. Dr Jaganathan Selva: Mehta Hospital, Chennai, India

## Author Contributions

**Conceptualization:** Bharath Kumar Tirupakuzhi Vijayaraghavan, Dilanthi Priyadarshini, Abi Beane, Rashan Haniffa.

**Data curation:** Bharath Kumar Tirupakuzhi Vijayaraghavan, Dilanthi Priyadarshini, Aasiyah Rashan, Abi Beane, Rashan Haniffa.

**Formal analysis:** Bharath Kumar Tirupakuzhi Vijayaraghavan, Dilanthi Priyadarshini, Aasiyah Rashan, Abi Beane, Rashan Haniffa.

**Funding acquisition:** Abi Beane, Rashan Haniffa.

**Methodology:** Dilanthi Priyadarshini, Aasiyah Rashan, Abi Beane, Ramesh Venkataraman, Nagarajan Ramakrishnan, Rashan Haniffa.

**Project administration:** Bharath Kumar Tirupakuzhi Vijayaraghavan, Rashan Haniffa.

**Supervision:** Bharath Kumar Tirupakuzhi Vijayaraghavan, Abi Beane, Ramesh Venkataraman, Nagarajan Ramakrishnan.

**Writing – original draft:** Bharath Kumar Tirupakuzhi Vijayaraghavan, Abi Beane, Ramesh Venkataraman, Nagarajan Ramakrishnan, Rashan Haniffa.

**Writing – review & editing:** Bharath Kumar Tirupakuzhi Vijayaraghavan, Dilanthi Priyadarshini, Abi Beane, Ramesh Venkataraman, Nagarajan Ramakrishnan, Rashan Haniffa.

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
