## [Decision Letter · Decision Letter 0]

14 Oct 2020

PONE-D-20-27684

Validation of a simplified risk prediction model using a cloud based critical care registry in a lower-middle income country

PLOS ONE

Dear Dr. Tirupakuzhi Vijayaraghavan,

Thank you for submitting your manuscript to PLOS ONE. After careful consideration, we feel that it has merit but does not fully meet PLOS ONE’s publication criteria as it currently stands. Therefore, we invite you to submit a revised version of the manuscript that addresses the points raised during the review process.

We look forward to receiving your revised manuscript.

Kind regards,

Aleksandar R. Zivkovic

Academic Editor

PLOS ONE

2. Please note that there appears to be a discrepancy in the number of participants, variously given as n=2094 and n=2062.

4.We note that you have indicated that data from this study are available upon request. PLOS only allows data to be available upon request if there are legal or ethical restrictions on sharing data publicly. For information on unacceptable data access restrictions, please see http://journals.plos.org/plosone/s/data-availability#loc-unacceptable-data-access-restrictions.

5.Thank you for stating the following in your Competing Interests section: 

[None ].

6. One of the noted authors is a group or consortium [IRIS collaborators.]. In addition to naming the author group, please list the individual authors and affiliations within this group in the acknowledgments section of your manuscript. Please also indicate clearly a lead author for this group along with a contact email address.

7. Your ethics statement should only appear in the Methods section of your manuscript. If your ethics statement is written in any section besides the Methods, please delete it from any other section.

8. Please ensure that you refer to Figure 1 in your text as, if accepted, production will need this reference to link the reader to the figure.

Reviewers' comments:

Reviewer #1: Congratulations for your research. This is an extensive work and the authors should be commended for this. This article will interest many readers. However, there are some minor comments in order to improve the presentation of this article.

- As excluded criteria of the study you report only the age and the length of stay in the ICU. Do you mean that in your dataset are included pregnant or burned patients and patients with the diagnosis of brain death or end-stage malignacy patients? The above case-mix are excluded from the majority of studies and scoring systems. It is not suprising then that the model has a poor calibration. The fact that the above patients are not excluded from the initial dataset is a disadvantage of the method. You should add a discussion about this in the manuscript.

- As you described in detail the AUROC values that indicate the descrimination, you should describe as well the values that indicate wheather the calibration is good or poor. (line 158)

-Line 224. Replace "By" with "by". Also in lines 223-227 there is a long sentence without any comma.

- Of course you need a further investigation in larger dataset in order to generalize the scoring system, but next time it would be better if you perform a prospective study, in order to minimize the missing data and to increase the accuracy of the validation.

Reviewer #2: This paper presents the validation of a new risk prediction model, termed e-Tropics, in a new data set. The results indicated that while the model discriminated well, the calibration was poor.

The methods used in the paper were valid and reasonable and the results potentially useful to users of e-TropICS however the paper lacked clarity in some areas and appeared to be missing information in others. The paper would benefit from additional analyses / more detailed consideration as detailed below.

Major comments:

It is not clear from the paper whether or not any of the institutions used in developing the model were included in the validation data. This is potentially important since case mixes may be different in different institutions. This important information should be carefully described for this data set (rather than stating as having been published elsewhere). The number of participants from each institution in this validation set should be included. If the validation set comprises a combination of institutions used and not used in the model building exercise, analysis containing each subset could be included in sensitivity analysis to show whether those subsets differ in terms of discrimination and calibration.

It would be useful if Table 1 contained summary information on the cohort from e-tropICS in an additional column - then readers can see where the lack of calibration may be arising from. Rather than use of all of the footnote symbols why not just say, for example: Mean heart rate (SD). Some abbreviations are missing definitions which should be included in the footnotes for the table. Why does Table 1 not include all of the e-tropICS variables too? Please update the table to include these. How do we interpret the summaries from Table 1? Obviously they show the difference between those who survived and otherwise but clinical interpretation would be nice here too - how sick is this cohort?

The lack of calibration could be an issue if it is in a region of the predictive distribution where it is important to calibrate well. It is important to know where the model is over or under-estimating the risk. As such, a calibration curve should be included in the paper to help to show readers where the lack of calibration is exhibited.

Sensitivity analysis should be used to show the discrimination and calibration of the data without imputation (i.e. complete case analysis). This will show how much imputation is impacting on the results.

The statistical analysis section describes application to multiple models ("each of the models'..." line 154), t-tests and chi-square tests. Other than the use of chi-squared in the HL C test, where are these tests performed? I don't see any results. It may be interesting to use such tests to compare the validation set with the e-tropICS though.

How to interpret the Brier score should be included in the statistical methods section and I note that the Brier score is not a particularly good measure of accuracy (https://diagnprognres.biomedcentral.com/articles/10.1186/s41512-017-0020-3).

In the multiple imputation section, predictive mean matching should not be used for categorical variables since it can generate non-integer results. For categorical variables one of the logit options should be used as described here: https://www.stata.com/manuals13/mimiimpute.pdf#mimiimpute. Table 2 should include all of the e-tropICS variables and the missingness for all of them. Categorical variables should show the levels so that we can see whether missingness relates to particular levels of a categorical variable. Line 202 of the results includes results not presented in Table 2. It is not immediately evident from the presentation of Table 2 what sorts of variables (categorical or numerical) these are.

Table 2 does not seem to contain all of the parameters described in the e-tropICS model 2 of Haniffa et al. (2017) and contains ones that should not be in the model. This is quite confusing and raises the question as to which model of Haniffa et al. was actually used.

Table 3 appears to be showing results that do not match the rest of the paper. Regarding sensitivity and specificity and cutpoints, it would be better to see a range of probability cutpoints together with the sensitivity so that the reader can see how different choices impact on results.

The ROC curve and estimate do not match the results and abstract.

The discussion sentence on lines 215-217 regarding the ability of "the model [to] identify those patients at greatest risk of death, but has less ability to differentiate between degrees of severity of illness." isn't shown by the results in their current state. Speculation about the case mix is similar. It would be good if you could show in the results that case mix differs and that there is an issue with calibration by severity of illness. The calibration curve and comparisons between the e-tropICS data and the validation data may help with that.

Minor comments:

Spaces are needed before the bracketing of abbreviations throughout, no need for repeating abbreviations, some abbreviations are not defined where they first appear (eg. abstract AUROC, HL C).

Why is the model described as Model 1 (line 109)? Are there other models that were meant to be presented also?

The e-tropICS model should be described as being internally validated rather than just "validated".

---

## [Author Response · Author response to Decision Letter 0]

16 Dec 2020

We would like to express our since gratitude to the Editorial team and Reviewers for their feedback and insights. We respond below:

Journal and Academic Editor comments:

Response: Changes made. 

2. Please note that there appears to be a discrepancy in the number of participants, variously given as n=2094 and n=2062.

Response: Apologies for the lack of clarity. We excluded 32 patients who did not meet inclusion criteria from 2094. This has been specified under the first paragraph of the ‘results’ and under the ‘results’ section of the abstract as well. 

Response: This current study is a registry-based analysis. Registries would essentially be untenable if individual patient consent was required. Each participating hospital and ICU in the registry decided in conjunction with their ethics committee what the consent model would be at their site. At some participating hospitals, this included a waiver and at some others a common written consent form at ICU admission authorizing the collection of deidentified data.We have now added these details into the Methods section (under ethics and consent). 

Response: Our agreement with participating sites in the registry is only for the sharing of deidentified data between them and the registry coordinating centre for the purposes of audit, quality improvement and specific research questions. We are not allowed to post data on a repository or any other public database. 

As stated under ‘availability of study data’ , we are committed to data sharing. Data will be made available to qualified researchers who provide a detailed and methodologically sound proposal with specific aims that are clearly outlined. Such proposals will be screened by the registry steering committee for approval. Data sharing will be for the purposes of medical research and under the auspices of the consent under which the data were originally gathered. 

To gain access, qualified researchers will need to sign a data sharing and access agreement and will need to confirm that data will only be used for the agreed upon purpose for which data access was granted. Researchers can contact the corresponding author (bharath@icuconsultants.com) through electronic mail for such access. 

[None ].

Response: Done

6. One of the noted authors is a group or consortium [IRIS collaborators.]. In addition to naming the author group, please list the individual authors and affiliations within this group in the acknowledgments section of your manuscript. Please also indicate clearly a lead author for this group along with a contact email address.

Response: Updated- highlighted under author information section. 

Lead (joint first )authors are Bharath Kumar Tirupakuzhi Vijayaraghavan and Dilanthi Priyadarshini.

Corresponding author is Bharath Kumar Tirupakuzhi Vijayaraghavan and email address is bharath@icuconsultants.com

7. Your ethics statement should only appear in the Methods section of your manuscript. If your ethics statement is written in any section besides the Methods, please delete it from any other section.

Done.

8. Please ensure that you refer to Figure 1 in your text as, if accepted, production will need this reference to link the reader to the figure.

Response: change made.

Response to reviewer comments: PLoS ONE

Reviewer 1:

Congratulations for your research. This is an extensive work and the authors should be commended for this. This article will interest many readers. However, there are some minor comments in order to improve the presentation of this article.

Response: Thank you. 

As excluded criteria of the study you report only the age and the length of stay in the ICU. Do you mean that in your dataset are included pregnant or burned patients and patients with the diagnosis of brain death or end-stage malignacy patients? The above case-mix are excluded from the majority of studies and scoring systems. It is not suprising then that the model has a poor calibration. The fact that the above patients are not excluded from the initial dataset is a disadvantage of the method. You should add a discussion about this in the manuscript.

Yes, our dataset includes pregnant patients and patients with malignancy (any stage). Only 6 patients coded as ‘burns’ are part of the dataset. Patients who are admitted with catastrophic brain injury who go on to become brain dead or are declared brain dead in the ICU are also part of the dataset. However, the overall proportion of all these patient groups is only 0.3% and unlikely to have impacted on calibration in our view.

As you described in detail the AUROC values that indicate the descrimination, you should describe as well the values that indicate wheather the calibration is good or poor. (line 158)

Response: Thank you. This has been added to the lines 165 and 166 and now reads higher values of the Hosmer-Lemeshow C-Statistic indicate poorer calibration

Line 224. Replace "By" with "by". Also in lines 223-227 there is a long sentence without any comma.

Done and corrections made.

Of course you need a further investigation in larger dataset in order to generalize the scoring system, but next time it would be better if you perform a prospective study, in order to minimize the missing data and to increase the accuracy of the validation.

Response: We thank the reviewer for this feedback and we will certainly incorporate this in our future study designs. 

Reviewer 2: 

This paper presents the validation of a new risk prediction model, termed e-Tropics, in a new data set. The results indicated that while the model discriminated well, the calibration was poor.

The methods used in the paper were valid and reasonable and the results potentially useful to users of e-TropICS however the paper lacked clarity in some areas and appeared to be missing information in others. The paper would benefit from additional analyses / more detailed consideration as detailed below.

Major comments:

It is not clear from the paper whether or not any of the institutions used in developing the model were included in the validation data. This is potentially important since case mixes may be different in different institutions. This important information should be carefully described for this data set (rather than stating as having been published elsewhere). The number of participants from each institution in this validation set should be included. If the validation set comprises a combination of institutions used and not used in the model building exercise, analysis containing each subset could be included in sensitivity analysis to show whether those subsets differ in terms of discrimination and calibration.

Response: We thank the reviewer for raising these important questions. None of the institutions involved in the development of the model were involved in validation. We have now clarified this in the Methods (section on study setting- lines 124-26).

The information on number of participants from each institution has now been added (supplementary table 3).

The model was developed on a completely different dataset and there is no overlap with the patients in the validation set. The validation dataset includes patients from 7 ICUs across India. 

It would be useful if Table 1 contained summary information on the cohort from e-tropICS in an additional column - then readers can see where the lack of calibration may be arising from. Rather than use of all of the footnote symbols why not just say, for example: Mean heart rate (SD). Some abbreviations are missing definitions which should be included in the footnotes for the table. Why does Table 1 not include all of the e-tropICS variables too? Please update the table to include these. How do we interpret the summaries from Table 1? Obviously they show the difference between those who survived and otherwise but clinical interpretation would be nice here too - how sick is this cohort?

Response: Our understanding is that the reviewer is seeking information on the baseline characteristics of the cohort from the original manuscript by Haniffa et al. We have now included this in Table 1. There are no missing variables in the table- e-TropICS includes variables published under ‘model 1’ of the original Haniffa manuscript. We apologize for the lack of clarity.

Changes made to abbreviations as suggested.

Table 1 presents the baseline characteristics of the entire cohort and by survival status. In terms of illness severity, approximately 1/4th of the cohort was ventilated on admission and approximately 1/5th were on vasopressors at admission. 

In addition, we have added a supplementary fig 1, which presents the commonest APACHE II diagnostic categories and the corresponding outcomes.

The lack of calibration could be an issue if it is in a region of the predictive distribution where it is important to calibrate well. It is important to know where the model is over or under-estimating the risk. As such, a calibration curve should be included in the paper to help to show readers where the lack of calibration is exhibited.

Response: We have now added this- Figure 2

Sensitivity analysis should be used to show the discrimination and calibration of the data without imputation (i.e. complete case analysis). This will show how much imputation is impacting on the results.

Response: We now provide the results of the CCA as well in our Results/Appendix section (supplementary table and figures 2). However, given that missingness is unlikely to be completely random (MCAR), we have chosen the imputation model as the primary analysis. 

The statistical analysis section describes application to multiple models ("each of the models'..." line 154), t-tests and chi-square tests. Other than the use of chi-squared in the HL C test, where are these tests performed? I don't see any results. It may be interesting to use such tests to compare the validation set with the e-tropICS though.

Response: Thank you. This has been corrected as one model

How to interpret the Brier score should be included in the statistical methods section and I note that the Brier score is not a particularly good measure of accuracy (https://diagnprognres.biomedcentral.com/articles/10.1186/s41512-017-0020-3).

Response: We accept the reviewer feedback and have deleted this metric. 

In the multiple imputation section, predictive mean matching should not be used for categorical variables since it can generate non-integer results. For categorical variables one of the logit options should be used as described here: https://www.stata.com/manuals13/mimiimpute.pdf#mimiimpute. 

Response: We apologise for the lack of clarity. The method we used was sequential imputation using chained equations , a multivariate approach that allows the flexibility of modelling different types of data within the same model with different rules being chosen based on type of data( PMM for continuous, logit for categorical and so on). https://www.stata.com/manuals13/mimiimputechained.pdf#mimiimputechained

Table 2 should include all of the e-tropICS variables and the missingness for all of them. Categorical variables should show the levels so that we can see whether missingness relates to particular levels of a categorical variable. Line 202 of the results includes results not presented in Table 2. It is not immediately evident from the presentation of Table 2 what sorts of variables (categorical or numerical) these are.

Response: We have made this change now. 

Table 2 does not seem to contain all of the parameters described in the e-tropICS model 2 of Haniffa et al. (2017) and contains ones that should not be in the model. This is quite confusing and raises the question as to which model of Haniffa et al. was actually used.

Response: The model used is ‘model 1’ from Haniffa et al’s paper. We apologize for the lack of clarity. 

Table 3 appears to be showing results that do not match the rest of the paper. Regarding sensitivity and specificity and cutpoints, it would be better to see a range of probability cutpoints together with the sensitivity so that the reader can see how different choices impact on results.

Response: We have addressed this now by adding a supplementary table (Suppl. Table 1) which shows a range of probability cutpoints with sensitivity and specificity. 

The ROC curve and estimate do not match the results and abstract.

Response: Corrected in both abstract and results- we apologise for this oversight. 

The discussion sentence on lines 215-217 regarding the ability of "the model [to] identify those patients at greatest risk of death, but has less ability to differentiate between degrees of severity of illness." isn't shown by the results in their current state. Speculation about the case mix is similar. It would be good if you could show in the results that case mix differs and that there is an issue with calibration by severity of illness. The calibration curve and comparisons between the e-tropICS data and the validation data may help with that.

Response: To address this, we have now included summary information from Haniffa et al.’s original manuscript into Table 1. We have also included the calibration curve. 

Minor comments:

Spaces are needed before the bracketing of abbreviations throughout, no need for repeating abbreviations, some abbreviations are not defined where they first appear (eg. abstract AUROC, HL C).

Response: Changes made. 

Why is the model described as Model 1 (line 109)? Are there other models that were meant to be presented also?

Response: In Haniffa et al. ‘s previous work, the variables included in e-TropICS were described as Model 1. This is the model we have validated. Haniffa et al. also developed a prognostic model called TropICS which we have not validated. 

The e-tropICS model should be described as being internally validated rather than just "validated".

Response: Our work is external validation ( on an entirely different dataset) and not internal validation. 

Figure 2: Calibration plot of the expected probability (deciles) versus the observed probability.

 Supplementary Figure 1: Commonest APACHE II diagnostic categories and ICU outcomes

Supplementary Table 1: Variation of sensitivity and specificity at different cutpoints

Cutpoint Sensitivity Specificity Correctly Classified

>=0.1 99.87% 4.01% 13.61%

>=0.2 88.60% 60.78% 63.57%

>=0.3 66.97% 74.83% 74.05%

>=0.4 59.97% 81.11% 78.99%

>=0.5 52.85% 85.71% 82.42%

>=0.6 48.70% 90.14% 85.99%

>=0.7 40.67% 94.10% 88.74%

>=0.8 23.19% 96.78% 89.41%

>=0.9 4.79% 98.79% 89.37%

---

## [Editor Report · Decision Letter 1]

21 Dec 2020

Validation of a simplified risk prediction model using a cloud based critical care registry in a lower-middle income country

PONE-D-20-27684R1

Dear Dr. Tirupakuzhi Vijayaraghavan,

We’re pleased to inform you that your manuscript has been judged scientifically suitable for publication and will be formally accepted for publication once it meets all outstanding technical requirements.

Kind regards,

Aleksandar R. Zivkovic

Academic Editor

PLOS ONE

---

## [Editor Report · Acceptance letter]

23 Dec 2020

PONE-D-20-27684R1 

Validation of a simplified risk prediction model using a cloud based critical care registry in a lower-middle income country 

Dear Dr. Tirupakuzhi Vijayaraghavan:

I'm pleased to inform you that your manuscript has been deemed suitable for publication in PLOS ONE. Congratulations! Your manuscript is now with our production department. 

Kind regards, 

on behalf of

Dr. Aleksandar R. Zivkovic 

Academic Editor

PLOS ONE